# Remodeling of Ion Channel Trafficking and Cardiac Arrhythmias

**DOI:** 10.3390/cells10092417

**Published:** 2021-09-14

**Authors:** Camille E. Blandin, Basile J. Gravez, Stéphane N. Hatem, Elise Balse

**Affiliations:** 1INSERM, Unité de Recherche sur les Maladies Cardiovasculaires, le Métabolisme et la Nutrition—UNITE 1166, Sorbonne Université, EQUIPE 3, F-75013 Paris, France; camille.blandin@sorbonne-universite.fr (C.E.B.); basile.gravez@sorbonne-universite.fr (B.J.G.); stephane.hatem@sorbonne-universite.fr (S.N.H.); 2ICAN—Institute of Cardiometabolism and Nutrition, Institute of Cardiology, Pitié-Salpêtrière Hospital, Sorbonne University, F-75013 Paris, France

**Keywords:** trafficking, ion channels, ion channel partners, arrhythmias

## Abstract

Both inherited and acquired cardiac arrhythmias are often associated with the abnormal functional expression of ion channels at the cellular level. The complex machinery that continuously traffics, anchors, organizes, and recycles ion channels at the plasma membrane of a cardiomyocyte appears to be a major source of channel dysfunction during cardiac arrhythmias. This has been well established with the discovery of mutations in the genes encoding several ion channels and ion channel partners during inherited cardiac arrhythmias. Fibrosis, altered myocyte contacts, and post-transcriptional protein changes are common factors that disorganize normal channel trafficking during acquired cardiac arrhythmias. Channel availability, described notably for hERG and K_V_1.5 channels, could be another potent arrhythmogenic mechanism. From this molecular knowledge on cardiac arrhythmias will emerge novel antiarrhythmic strategies.

## 1. Introduction

Cardiac arrhythmias are the main cause of strokes, heart failure, or sudden death and are a major health problem. Alteration in the functional activity of ion channels is central to most inherited and acquired cardiac arrhythmias. However, the underlying mechanisms are still incompletely understood. One reason is the extreme complexity of the processes involving a myriad of proteins, referred to as trafficking, that are a major source of arrhythmogenic mechanisms.

Studies on the genetics of inherited cardiac arrhythmias show that certain loss-of-function (LOF) mutations are associated with altered channel trafficking that are retained in intracellular organelles. Then, a number of mutations have been identified in several protein partners that chaperone channels or tether or anchor channels to the microdomains of the sarcolemma of cardiomyocytes. These studies have firmly established the role of alteration in channel trafficking in cardiac arrhythmias. Myocardial remodeling associated with cardiopathies and heart failure, the main risk factor of cardiac arrhythmias, is also associated with default of channel trafficking, notably, due to changes in microenvironment of cardiomyocytes. Things are even more complex given the number of remote factors, for instance, the membrane lipid composition or the metabolic phenotype of cardiomyocytes, that can regulate channel trafficking and hence contribute to arrhythmogenic processes.

In this article, first an initial update on knowledge on the physiology of channel trafficking will be provided and then evidence on trafficking defects obtained during inherited and acquired cardiac arrhythmias will be reviewed. Note that in the general concept of trafficking, we will include intracellular vesicular trafficking (i.e., surface regulation) and addressing in specialized areas of the myocyte (i.e., local regulation).

## 2. The Trafficking Machinery of Cardiomyocytes

Cardiac excitability is generated exclusively by cardiomyocytes. Some of them are able to depolarize spontaneously the pacemaker cells of the sinus node, some others are specialized in the transmission of the electrical impulse (Purkinje cells), whereas the vast majority are involved in the excitation–contraction coupling process. In cardiomyocytes, electrogenic systems are composed mainly of voltage-dependent channels (VOC), some pumps, and exchangers. At rest, cardiomyocytes are polarized around −90 mV, the equilibrium potential of K+, which is clamped by the background inward rectifier potassium current, IK1, and the Na+-K+ ATPase pump. The fast upstroke of the action potential (AP) is caused by the activation of a large sodium current due to the opening of the Nav1.5 sodium channel. The second phase of the AP is a plateau phase lasting a few hundreds of msec, during which the excitation–contraction coupling process is activated, which is controlled by the L-type calcium current. A T-type calcium current is recorded in cardiomyocytes, but its role is not established. A number of potassium currents shape the AP; they belong to voltage-dependent and inward rectifier potassium channel families. There are also two pore channels that are voltage gated but recruited in response to various stimuli, such as pH. In addition to a change in the membrane electrical potential, the continuous variations in free intracellular calcium during the EC coupling process are a master regulator of cardiac excitability mainly via the electrogenic Na+-Ca+ exchanger (NCX); the direct effect of calcium on channel gating (rapid inactivation of the L-type calcium channel); or the activation of a second messenger, such as CaMK-II. There are differences between atrial and ventricular myocytes that are mainly due to the cellular architecture and notably the organization of the t-tubular system, which is highly developed in ventricular myocytes (see for more information [1]).

### 2.1. Molecular Basis of Ion Channel Trafficking

Ion channels, such as secreted proteins, are synthesized in the endoplasmic reticulum (ER) by ER-bound ribosomes. The ER lumen constitutes the future extracellular domain of the channel and permits cellular quality control to ensure that only correctly folded channels can exit the compartment [2]. Misfolded or misassembled channels exhibit ER-retention signals, the best known of which are the RXR and KDEL motifs [3,4]. If misfolded or misassembled, the channels will be retained in the ER and the unfolded protein response (UPR) will be triggered using three main pathways: PERK, ATF6, and IRE1 [5,6]. The latter two activate a subset of target genes involved in the ER-associated degradation system (ERAD). The ERAD induces cytosolic degradation in the proteasome of misfolded proteins after ubiquitylation by ER-bound E3-ubiquitin ligases [7]. Specific motifs are also involved in active transport out of the ER, such as the diacidic DAD or non-acidic VXXSL and VXXSN motifs [8]. The transport of channels between the ER and the Golgi complex (GC) takes place via coated transition vesicles: COPII-coated vesicles move toward the GC and COPI-coated vesicles return to the ER [9]. After exiting the ER, the channels are transported via vesicles until they reach the plasma membrane. Transportation is achieved by the fusion of a donor compartment with an acceptor compartment. Each fusion step involves SNARE proteins (soluble N-ethylmaleimide-sensitive factor attachment protein receptors) carried by both the donor vesicle membrane (one vesicle-membrane SNARE) and the acceptor membrane (three target-membrane SNAREs), which destabilize the lipid bilayers and allow the channel to move to the acceptor compartment when interacting [10]. Ion channel trafficking in the cytoplasm is coordinated by various Rab GTPases, monomeric GTPases of the Ras superfamily. Interestingly, these GTPases can serve as markers of the progression of intracellular trafficking in the antero- and retrograde directions since they are associated with specific vesicles. For instance, Rab8 is involved in the delivery of newly synthetized channels and associated with the secretory vesicle. Rab4, Rab5, Rab7, Rab9, and Rab11 are associated with vesicles following ion channel internalization. Rab4 and Rab5 are associated with the early endosome and involved mainly in fast recycling back to the plasma membrane [11,12]. Rab11 is associated with the recycling endosome and essential to a slower recycling pathway [13,14]. Rab7 and Rab9 are associated with the late endosome and direct the trafficking toward the lysosome [15,16] or the proteasome [17] for degradation. All of these Rab GTPases are expressed in cardiomyocytes. If the vesicles transporting ion channels can progress through the cell, it is because molecular motors exist that allow their transport along the cytoskeleton in opposite directions. The vesicles travel in the anterograde direction along the microtubules by the action of different kinesins and thereafter along the actin cytoskeleton by the action of myosin V. In the retrograde direction, the molecular motors are dynein (mostly dynein I) and myosin VI, associated with microtubules and actin, respectively [18]. Figure 1 illustrates the main fundamental mechanisms responsible for ion channel trafficking in cardiomyocytes after Golgi exit or after internalization.

Another group of proteins central to the trafficking of cardiac ion channels belongs to the membrane-associated GUanylate kinase (MAGUK) family. These proteins are characterized by the presence of multiple protein–protein interaction domains, including PDZ (post synaptic density protein (PSD-95), Drosophila disc large tumor suppressor (Dlg1), and zonula occludens-1 protein (zo-1)) and SH3 domains (SRC Homology 3 Domain), explaining how they anchor and tether macromolecular complexes to the plasma membrane. Cardiomyocytes express several MAGUK proteins, such as PSD-95 (post synaptic density protein), chapsyn-110, synapse-associated protein 102, SAP97 (synapse-associated protein 97), and CASK (calcium/calmodulin dependent serine protein kinase). SAP97, the most well-characterized MAGUK protein in the myocardium, interacts with several ion channel families: K_V_1.5 [19,20], K_V_4.x [21,22], Kir2.x [23,24,25], and Na_V_1.5 [24,26]. Apart from anchoring the ion channels to the plasma membrane microdomains, SAP97 is involved in the trafficking machinery. It regulates the formation of Kir2.1/Na_V_1.5 complexes, the two critical channels underlying *I*_K1_ and *I*_Na_, responsible for the maintenance of the resting membrane potential and the rapid depolarization during the upstroke of the AP, respectively [24]. This specific multi-channel organization enables reciprocal modulation. Interestingly, when Na_V_1.5 is overexpressed, internalization of Kir2.1 appears to be reduced, suggesting that the channels regulate each other in part through retrograde trafficking [24]. Recently characterized in the myocardium, the MAGUK protein CASK negatively regulates the surface expression of Na_V_1.5 by impeding the Golgi-to-plasma-membrane trafficking of the channel [27,28] and reduces the Kir2.x-mediated *I*_K1_ current (personal unpublished data). Of note, previous publications have reported that SAP97 and CASK act synergistically to regulate the trafficking and targeting of Kir2.2 potassium channels at the basolateral membrane of epithelial cells [23]. A macromolecular complex composed of a Mint1/Veli/SAP97/CASK and Kir2.x channel has been characterized in heart and skeletal muscle cells. This supramolecular complex also interacts with components of the dystrophin–glycoprotein complex, including syntrophin, dystrophin, and dystrobrevin, at the neuromuscular junction, although with lower affinities [29]. Therefore, both SAP97 and CASK appear to be central proteins of a macromolecular complex that participate in the trafficking and plasma membrane localization of ion channels.

### 2.2. Why Ion Channel Trafficking Is Crucial for Normal Cardiac Electrical Properties

The trafficking machinery continuously routes, delivers, and recycles electrogenic proteins, such as channels, pumps, and exchangers, into microdomains of the plasma membrane of cardiomyocytes. Distinct microdomains can be identified, t-tubules, lateral membrane (LM), and interacted disk (ID), all having precise physiological roles: the excitation–contraction coupling process, propagation of the action potential, electromechanical coupling between myocytes, or mechano-transduction processes. Such ultrastructural organization of cardiomyocytes is crucial for the functional polarization of cardiomyocytes and for the anisotropic transmission of the electromechanical wave at the tissue level.

The molecular organization, including trafficking processes, has been investigated in great detail for the gap junction channel complex, the connexosome, at the ID. Less known is the role of the sodium channel in functional cardiac polarization, although Na_V_1.5 trafficking and organization appears increasingly central to the genesis of cardiac arrhythmias, as described below. While Na_V_1.5 is highly concentrated at the ID, Na_V_1.5 channels at the LM show a lower density (3- to 8-fold smaller current) [30,31]. From this perspective, membrane microdomain distribution of Na_V_1.5 has been further studied using high-resolution microscopy techniques. Scanning ion-conductance microscopy identified three topographic structures at the LM: t-tubules, crest, and z-grooves corresponding to t-tubule openings [32]. High-resolution patch clamp recordings showed that Na_V_1.5 channels are not distributed homogeneously along the LM. Na_V_1.5 channels are organized in clusters, the largest being located around crests (probably costamere regions of the LM) and the smaller ones at t-tubules. Moreover, using multicolor single-molecule localization microscopy, affinity between LM Na_V_1.5 clusters and the t-tubule marker BIN1 was observed, indicating that a reasonable subset of intracellular Na_V_1.5 is expressed at the t-tubule membrane [33]. Several partners of Na_V_1.5 were identified, such as connexin-43 (in gap junctions), plakophilin-2 (in desmosomes), ankyrin-G (involved in actin cytoskeleton binding), the MAGUK protein SAP97 (in the ID), and syntrophin (at the LM). All these partners exert a positive regulatory effect on *I*_Na_ since their silencing in vitro, knock out in vivo, or reduced expression in the Duchenne muscular dystrophy mouse model decrease both sodium current and Na_V_1.5 cell surface localization (for a review, see [34]). However, whether these proteins are involved in the trafficking of Na_V_1.5 (either anterograde or retrograde) or associated with the channel once inserted in the plasma membrane to stabilize the channel is not well defined. Among these partners, the newly characterized MAGUK protein CASK seems directly involved in the anterograde trafficking of the Na_V_1.5 channel in cardiomyocytes. As a syntrophin, CASK is restricted to the LM in the costameric dystrophin–glycoprotein complex and interacts with the Na_V_1.5 channel through its *C*-terminal PDZ-binding motif SIV [27]. In contrast to other Na_V_1.5 partners, CASK negatively regulates the *I*_Na_ current by impeding the early forward trafficking of Na_V_1.5 to the LM of cardiomyocytes [27,28]. Given its multi-modular structure, CASK can control channel delivery at adhesion points in cardiomyocytes, linking channel expression to structural organization of cardiomyocytes [28].

Additional studies are necessary to identify mechanisms that govern the precise addressing and organization of channels into specialized domains of the plasma membrane of cardiomyocytes. In this context, real-time study of channel and partner (co-)trafficking should help understand whether interactions with and/or competition between different channel partners during intracellular trafficking steps direct the final destination of ion channels. Recently, using cutting-edge live-imaging technologies, the group of Tseng showed that KCNQ1 and KCNE1 traffic through separate routes and only assemble at the sarcolemma to form *I*_Ks_ [35]. Furthermore, to understand the development of arrhythmias associated upstream or downstream of myocardial remodeling, the question of the interplay between electrical and structural polarization must be addressed.

## 3. Tribute to Inherited Cardiac Arrhythmias for Knowledge on Channel Trafficking

In the late 1990s, the discovery that LOF mutations are responsible for trafficking defects whereas biosynthesis and biophysical properties of channels are not altered is the first demonstration of the role played by abnormal channel trafficking in the genesis of cardiac arrhythmias. Thereafter, more than 230 articles associating inherited arrhythmias with trafficking defects have been published establishing firmly that channel trafficking is indeed a major component of cardiac arrhythmias.

Most trafficking defect mutations resulting in LOF involve endoplasmic reticulum (ER) exit defects, leading to the targeted degradation of misfolded channels by the ERAD system. Historically, these mutations were first identified as associated with long QT syndrome and involving *HERG* [36] (LQT2) and then *KCNQ1* [37] (LQT1) or associated with Brugada syndrome and carried by *SCN5A* [38]. Gain-of-function (GOF) mutations supported by enhanced channel trafficking have also been reported for *KCNQ1* in both inherited atrial fibrillation and short QT syndrome [39,40] as well as for *KCND3* in atrial fibrillation (AF) [41]. Although less documented, other trafficking stages can also be impacted. GOF mutations have also been identified for TRPM4, which encodes a nonselective cation channel, in human progressive familial heart block type I, and involve a defect in endocytosis [42]. In addition to the mutations in ion channels leading to arrhythmia, some mutations have been identified in the protein partners with which these channels and transporters associate and by which they are regulated. Next, we will summarize mutations in channel partners leading to trafficking defects in the context of most documented inherited cardiac arrhythmias, LTQ, and Brugada syndromes.

### 3.1. Long QT Syndrome and Ion Channel Partners

Long QT syndrome (LQTS) is a rare congenital syndrome (1:2500–1:10,000) characterized by prolonged QT intervals and associated with a risk of ventricular tachycardia, termed torsades de pointes (TdP) and cardiac sudden death [43]. Historically, congenital LQTS have been associated with multiple penetrant mutations in genes encoding the pore-forming α-subunit of ion channels, with a majority of mutations located in the *KCNQ1*, *KCNH2*, or *SCN5A* gene and responsible for LQT1-3 [44]. Recently, helped by advances in molecular biology and sequencing technology, important observations have been made regarding the development of LQTS and, more specifically, its association with ion channel partners.

#### 3.1.1. Long QT 4 (Ankyrin Mutations)

Encoded by three genes (*ANK-1-3*), ankyrins are expressed in many tissues [45]. As cytoskeleton-binding proteins, ankyrins anchor ion channels to the actin and spectrin cytoskeleton. A role of ankyrin-B (*ANK2*) in cardiac arrhythmias was first evidenced by the identification of a common mutation (E1425G) in patients from a French kindred presenting with atypical LQTS [46], initially referred to as LQT4 and then renamed ankyrin-B syndrome. The mutation E1425G reduces the targeting of Na/K ATPase, Na/Ca exchanger (NCX), and inositol-1, 4, 5-triphosphate receptors (InsP_3_R) to cardiomyocyte t-tubules, as shown in vitro in *AnkB*^+/−^-isolated adult cardiomyocytes, whereas no changes are observed at the transcription level. Ankyrin-B mutation in both human and heterozygous *AnkB*^+/−^ mice is responsible for various ECG abnormalities (bradycardia, heart rate variability), leading to severe arrhythmias and in some cases sudden death. Thereafter, beyond LQT, several loss-of-function ankyrin-B mutations were identified in patients with various cardiac arrhythmias notably associated with abnormal calcium homeostasis [47,48,49,50] and during AF [51].

#### 3.1.2. Long QT 9 (Caveolin-3 Mutations)

Caveolins 1–3 are specific membrane proteins that form a hairpin loop inside the membrane, keeping *C*- and *N*-terminals oriented toward the cytosol. Caveolins are the main constituents of caveolae, small compartments of the plasma membrane, which were first identified by electron microscopy [52]. These bulb-shaped invaginations connected to the plasma membrane by a collar are involved in many cellular functions [53]. First described as a major actor in endocytosis, they are also scaffolding proteins able to guide and tether signaling molecules and ion channels to the peri-membrane region [54]. Recently, evidence of a mechanical-stress regulation has emerged, with caveolae involved in mechano-adaptation and cellular homeostasis [55]. *CAV3*-encoded caveolin-3 is the main caveolin isoform expressed in the myocardium.

The regulation of cardiac *I*_Na_ by caveolin was first demonstrated by Yarbrough and co-workers. G-protein activation following β-adrenergic stimulation in rat cardiomyocytes leads to the opening of caveolae and probably the subsequent release of Na_V_1.5, indicating that caveolae could be a reservoir for sodium channel Na_V_1.5 [56]. Robust evidence of the role of caveolin-3 in long QT cardiac arrhythmias came from the identification of four mutations (F97C, S141R, T78M, and A85T) in patients referred for LQTS. Despite the fact that *CAV3* mutation results in late *I*_Na_ increase and that caveolin-3/Na_V_1.5 association is not disrupted, no evidence was reported as to whether trafficking was involved in the effect [57]. Indeed, caveolae also serve to organize and regulate signaling pathways. In this regard, Cheng and co-workers demonstrated that the co-expression of *SCN5A*, *SNTA1*, nNOS, and mutated *CAV3* (F97C) leads to increased N-nitrosylation of Na_v_1.5. nNOS inhibition reversed both the *CAV3*-F97C-dependent late *I*_Na_ and S-nitrosylation of Na_V_1.5 [58]. In addition, three distinct *CAV3* mutations (V14L, T78M, and L79R) affecting late *I*_Na_ in a similar way were identified in sudden infant death syndrome (SIDS) [59]. More recently, other channels have been implicated in action potential duration (APD) prolongation in association with *CAV3* mutations, such as Ca_V_1.2 and K_V_4.x channels. For instance, the *CAV3*-S141R mutation was shown to increase *I*_Ca,L_ density without affecting channel gating properties and to decrease *I*_to_ density and slow its activation properties [60]. The potassium inward rectifier channel Kir2.1 also associates with caveolin-3. *CAV3* mutations F97C, P104L, and T78M significantly decrease *I*_K1_ by reducing cell surface expression of the corresponding channel [61]. Altogether, these studies suggest that caveloae could be involved both in channel trafficking (i.e., rapid delivery of submembrane pools of ion channels) and scaffolding of macromolecular complexes, allowing ion channel regulation.

### 3.2. Brugada Syndrome and Ion Channel Partners

The Brugada syndrome (BrS) is another cause of life-threatening ventricular arrhythmias, characterized by repolarization abnormalities recorded in the right precordial leads of the electrocardiogram. BrS is an oligogenic/polygenic disease often characterized by alterations in sodium current properties, notably in the ventricular output track. However, the underlying molecular mechanisms of sodium current abnormalities are still being investigated and often involve default of channel trafficking.

#### 3.2.1. SCN1-3B Mutations

In addition to the pore forming α-subunit, four additional β-subunits encoded by four distinct genes (*SCN1B* (encoding the β1 and β1b isoforms) and *SCN2B*, *SCN3B*, and *SCN4B* (encoding the β2, β3, and β4 isoforms, respectively)) constitute the sodium channel. Na_V_1.5 β-subunits were identified as *I*_Na_ modulators because they increased Na_V_1.5 expression at the cell surface [62,63] and potassium currents by modifying the molecular actors’ gating properties of *I*_to_ and *I*_K1_ [64]. In addition, Na_V_-β are involved in cell adhesions by recruiting ankyrin-G to cell–cell contact sites [62].

The involvement of the β-subunit in the context of cardiac arrhythmia was first highlighted by Watanabe and colleagues with the identification of a mutation (W179X) in *SCN1B* in a patient presenting with the hallmarks of BrS [63]. In vitro, this mutation prevents the increase of current density observed when Na_V_1.5 is co-expressed with WT β1B [63]. Then, Holst and co-workers identified two additional mutations in *SCN1B* (H162P and R214Q) after screening BrS patients [65,66]. Electrophysiological measurements revealed that R214Q mutation is responsible for both outward current *I*_to_ increase and inward current *I*_Na_ reduction. Plus, Na_V_-β1b was found to directly interact with both Na_V_1.5 and K_V_4.3 [67]. Functional characterization of the H162P mutant indicates that hNav1.5/hNavβ1b-H162P association results in reduced peak *I*_Na_ current density [68]. Recently, a new *SCN1B* variant (A197V) was associated with BrS [69]. In addition to reducing *I*_Na_ current density, the Na_V_-β1-A197V mutant impairs channel trafficking to the plasma membrane, by retaining Na_V_1.5 channels in the intracellular compartment [70].

In parallel, Hu and co-workers identified a first missense mutation in the *SCN3B* gene (L10P) in a patient displaying Type-1 ST segment elevation. As the majority of mutations identified in *SCN1B*, *SCN3B*-L10P associated with reduced *I*_Na_ density. Confocal microscopy reveals a trafficking defect of the Na_V_1.5 channel, that it remains blocked in intracellular organelles instead of being tethered to the cell surface [71]. An *SCN3B*-V110I mutation carried by unrelated BrS patients was also identified as a common cause of BrS. This mutation also reduces *I*_Na_ by impairing forward trafficking [72]. A *SCN2B* missense mutation includes *SCN2B* in the list of candidate genes for BrS. Indeed, D211G mutation was identified in a patient presenting with clinical manifestations of BrS and was associated with reduced Na_v_1.5 channel cell surface expression [73]. As of now, *SCN4B* has not been associated with BrS. Apart from BrS, mutations in *SCN1B* were also linked to LQTS [74] and SIDS [67], suggesting a common mechanism altering cardiac electric activity, possibly ion channel trafficking.

#### 3.2.2. Ankyrin-G

Ankyrin-G (*ANK3*) is located at the ID and in the t-tubules of adult myocytes. The first piece of evidence of the role of ankyrin-G in ion channel trafficking in cardiomyocyte came from the identification of a mutation (E1053K) in *SCN5A* of a BrS patient [48]. Specially localized in the ankyrin-binding motif of Na_V_1.5, this mutation prevents Na_V_1.5 from binding to ankyrin-G, therefore significantly reducing its accumulation at the cell surface (ID, t-tubules). Interestingly, Golgi-mediated forward trafficking was not impaired, suggesting that Na_V_1.5 may bypass the conventional secretory pathway. Ankyrin-G silencing in adult cardiomyocyte in vitro reduces *I*_Na_ density without changing ion channel biophysical properties [75]. In addition, loss of Na_V_1.5–ankyrin-G interaction induces Na_V_1.5 retention in perinuclear regions and reduces Na_V_1.5 surface expression, notably at the ID [75]. This study suggests that ion channel trafficking is tightly linked to membrane micro-domains in adult cardiomyocyte. However, whether ankyrin-G is required for anterograde transport of the channel or for stabilization/retention of the channel at the surface is not clear. Ankyrin-G silencing also impairs electrical coupling by reducing connexin-43 (Cx43) expression and perturbs Na_V_1.5 localization at the ID [76]. Furthermore, as Cx43 seems crucial for microtubular network stability [77], Cx43 would be required for the formation of functional sodium channel complexes at the ID [78]. In vivo experiments using a cardiac-specific ankyrin-G KO mouse revealed decreased Na_V_1.5 ID population and reduced *I*_Na_, associated with the reorganization of the ID structure, as plakophilin-2 redistributes to the cytosolic regions [79]. Recently, Yang and co-workers revealed that ankyrin-G regulates the targeting of both Na_V_1.5 and K_ATP_ channels to ID but not to LM. Using quantitative STORM imaging, they demonstrated the existence of a Na_v_1.5/Kir6.2/AnkG complex at the ID. This study suggests a functional coupling between those two channels, with a common regulatory process implicating ankyrin-G [80].

#### 3.2.3. Other Trafficking Partner Mutations in Brugada Syndrome

Five cases of single amino acid substitutions of plakophilin 2 (PKP2) have been reported in patients with BrS without mutations in the BrS-related genes (*SCN5A*, *CACNA1C*, *GPD1L*, and *MOG1*) or sign of ARVC. Several in vitro data obtained from hiPS-derived cardiomyocytes of a patient with a *PKP2* mutation indicate that the mutated plakophilin 2 fails to target Na_V_1.5 channels at the plasma membrane [81]. Studying a human sample of the genetically elusive LQTS and BrS, Musa and co-workers provided the first piece of evidence involving a GOF missense mutation in *DLG1* (*SAP97*-M82T) favoring K_v_4.3-mediated *I*_to_ [82]. In this context, the SIV domain of the sodium channel has been shown to be responsible for the interaction with both SAP97 and CASK and a V2016M mutation in this SIV domain has been identified in a patient suffering from BrS [83].

## 4. Can Acquired Cardiac Arrhythmias Be Caused by Default of Channel Trafficking?

Acquired cardiac arrhythmias such as AF, scar-, or heart failure-related ventricular arrhythmias are often underlined by a complex substrate that includes fibrosis, neuro-hormonal activation, and post-translational changes in myriads of proteins. Consequently, a combination of mechanisms operates together to generate an arrhythmia such as abnormal pace making, triggered activities, local conduction block, and circuit of reentries of the electrical influx. Interestingly, despite this complexity, common pathophysiological pathways emerge that lead to default of channel trafficking and disorganization of the macromolecular complex. Alteration in myocyte–myocyte coupling, disorganization of membrane microdomains, and changes in the channel microenvironment appear as major mechanisms responsible for default of channel trafficking during acquired cardiac arrhythmias.

### 4.1. When Tissue Remodeling Disorganizes Cardiac Channel Trafficking

The delocalization of gap junctional channels, the connexins, from the intercalated disk (ID) to the LM of cardiomyocytes has been consistently reported during hypertrophic and fibrotic myocardial remodeling at both atrial and ventricular levels [84,85,86,87]. This connexin disorganization can reduce myocyte–myocyte electrical coupling and favor electrical conduction block and, hence, cardiac arrhythmias. This phenomenon is probably the first indication of default of channel trafficking during acquired cardiac arrhythmias. The remodeling of the extracellular matrix surrounding cardiomyocytes together with the abnormal coupling between myocyte and (myo-)fibroblasts favors gap junction channel disorganization. This has been demonstrated in vitro by atrial myocytes with a fibroblast co-culture that reproduce the delocalization of connexin at the LM of cardiomyocyte without gap junction organization and reduce myocyte–myocyte electrical coupling [88]. Delocalized Cx43 at the LM are predominantly in a dephosphorylated state, which has been shown elsewhere to contribute to abnormal targeting and assembly of connexins into gap junctions [84,85,89]. Another piece of evidence of the critical role of Cx43 in the normal electrical function of the mammalian heart has been provided by the study of the internally translated Cx43 isoform GJA1-20k. This auxiliary subunit contributes to the normal trafficking of Cx43, and its mutation M213L is associated with abnormal electrocardiograms and sudden death in mice models [89].

Several studies suggest that the disorganization of membrane microdomains containing the subpopulations of Na_V_1.5 channels could also contribute to the substrate of acquired cardiac arrhythmias, for instance, the direct targeting of hemi-channels to the ID via the microtubule plus-end-tracking protein EB1 [90] and the targeting of Ca_V_1.2 to t-tubules by BIN1 (bridging integrator 1), a protein involved in membrane invagination and endocytotic processes [91]. Notably, mice with cardiac BIN deletion present decreased t-tubule folding, resulting in the free diffusion of local extracellular calcium and potassium ions, lengthening of action potential, and abnormal susceptibility to ventricular arrhythmias [92]. Furthermore, the MAGUK protein CASK, which negatively regulates *I*_Na_ by impeding Na_V_1.5 anterograde trafficking to the LM, is downregulated in diseased atria of patients with AF or valve regurgitation and could contribute to the loss of anisotropy of the diseased atrial myocardium [27]. Of note, CASK expression is enhanced in the hypertrophied myocardia of patients suffering from aortic stenosis and heart failure with preserved ejection fraction. However, in this condition, the deleterious effect of CASK on ventricle pump function and cardiac arrhythmias has been attributed to its CaMKII activity and not to channel trafficking [93].

### 4.2. Keep Ion Channel Trafficking Balanced

As mentioned above, a fine balance between exocytosis and endocytosis of ion channels is necessary for normal cardiac electrical activity (for review, see [94]). In some conditions, the dysregulation of this balance can be arrhythmogenic (Figure 2).

#### 4.2.1. hERG, Channel at Demand

The *I*_Kr_ current is encoded by the human ether-a-go-go-related gene (*HERG* or *KCNH2*) in the heart [95,96]. *HERG* mutations resulting in reduced *I*_Kr_ cause type 2 long QT syndrome (LQT2), which predisposes individuals to life-threatening arrhythmias. The vast majority of *HERG* mutations are LOF mutations resulting from disrupted forward trafficking of the hERG channel and resulting in decreased *I*_Kr_ [97]. The hERG channel is also a notorious target for several classes of drugs that engender acquired LQTS [98]. For instance, apart from exerting a pore-block effect, tricyclic antidepressants, such as desipramine, were shown to increase hERG endocytosis and degradation following channel ubiquitination and simultaneously to inhibit hERG forward trafficking from the ER [99]. LQTS and *torsades de pointes* are known to be exacerbated by hypokalemia, with a moderate increase in serum [K^+^] capable of correcting LQTS in some patients [100]. Hypokalemia is, therefore, considered a risk factor for LQTS and sudden cardiac death. Guo and colleagues provided a clue to the role of extracellular potassium in the functional expression of the hERG channel [101]. In a rabbit model of hypokalemia, the QT interval prolongation correlates with a prolongation of APD90. In vitro, exposure to a potassium-depleted medium completely but reversibly eliminates the *I*_Kr_ current without affecting other potassium currents. The decrease in *I*_Kr_ is explained by a reduction in the sarcolemmal expression of the hERG channel, combined with increased internalization and lysosomal degradation of the channel following its ubiquitylation [101] (Figure 2A, left). These results provide a potential mechanism for hypokalemia-induced exacerbation of LQTS. Prolongation of the QT interval is also a prominent electrical disorder in patients suffering from diabetes mellitus. Shi and co-workers reported that cells incubated in a high-glucose medium display reduced hERG current and hERG protein expression of the mature (glycosylated) and immature (unglycolsylated) forms [102]. This reduced expression level is associated with impaired anterograde trafficking through reduced interaction between hERG and the chaperone protein Hsp90, leading to the activation of the UPR and subsequent hERG degradation [102] (Figure 2A, right).

#### 4.2.2. The Restless K_V_1.5 Channel

The other ion channel the functional expression of which is highly regulated by trafficking process is the voltage-gated K_V_1.5 channel. K_V_1.5 continuously oscillates between a mobile/trafficking state and a more static one, determining its level of expression at the sarcolemma [103]. The K_V_1.5 channel underlies an important repolarizing current of the plateau phase of the atrial myocytes, *I*_Kur_ [104]. Moreover *I*_Kur_ has been implicated in the electrical remodeling associated with the occurrence of AF [105,106], leading to the unmet expectation of K_V_1.5 blockers as specific anti-arrhythmic agents against AF. Nevertheless, dissection of the recycling process of the K_V_1.5 channel in normal and diseased atria has led to an important breakthrough in the pathophysiology of cardiac electrical remodeling.

a.K_V_1.5 channel trafficking needs fat

The plasma membrane of a cardiomyocyte is composed of various types of lipids, which determine the structural and physical properties of the membrane [107]. Cholesterol affects membrane fluidity, defined as the degree of ordering and motional freedom of a lipid-soluble molecule within a cellular membrane. Cholesterol is not randomly distributed in the plasma membrane but packed in a cholesterol-enriched domain called lipid rafts, which are dynamic platforms important for the delivery of proteins to the membrane and for sequestering proteins in close physical proximity to control their functional interactions [108]. Several cardiac ion channels are localized into a lipid raft, notably the caveolae containing the anchoring protein caveolin-3, which regulates their trafficking at the plasma membrane [94,109].

Another way by which cholesterol can influence membrane cardiac ionic channel function is by affecting its delivery at the plasma membrane. Depletion of membrane cholesterol using methyl-β-cyclodextrin (MβCD) increases the number of functional K_V_1.5 channels in atrial myocytes rapidly (in a few minutes), as shown by single K_V_1.5 channel activity recordings in cell-attached patch clamp configuration and fluorescence recovery after photobleaching (FRAP) experiments [110]. This increased functional expression is due to the exocytosis of channels from the recycling pathway, specifically the Rab11-coupled recycling endosome, and not from the insertion of neo-synthesized, membrane-addressed channels through the conventional secretory pathway. Indeed, the K_V_1.5 channel packed into the Rab11 recycling endosome, considered as a slow route for ion channel recycling [13,14], dissociates rapidly upon cholesterol depletion in atrial myocytes (Figure 2B). In addition to the K_V_1.5 channel, the *I*_Ks_ current appears to be regulated by intracellular trafficking. Indeed, chronic activation of calcium-dependent PKC (cPKC) induces a decrease in KCNQ1 membrane expression and *I*_Ks_ current via an increase in channel endocytosis dependent on dynamin II and Rab5 [111]. Interestingly, the use of statins restores the surface localization of the channel and its function by inhibiting the Rab5-associated early endosome [111,112]. Thus, anti-cholesterol therapies could be beneficial in diseases associated with chronic PKC activation and/or calcium handling defects, such as heart failure.

How is the membrane cholesterol regulation of cardiac channel turnover linked to cardiac arrhythmias? Growing evidence indicates a tight relationship between AF and metabolic disorders, such as obesity. Furthermore, experimental models of obesity and a high fat diet are characterized by an abnormal susceptibility to AF [113]. Several factors can explain this association between obesity, diet, and AF, and among them, a direct effect of a cholesterol diet on membrane composition and excitability is a probability [109,114].

b.Do not stress the K_V_1.5 channel

Cardiomyocytes are continuously submitted to mechanical forces during cardiac cycles that can be distinguished into stretch, shear stress, and strain constraints. For instance, shear stress is generated by the sliding of myocardial layers [115,116] and it can activate intracellular calcium transients [117,118], increase the beating rate of neonatal ventricular myocytes [119], and trigger propagating action potential (AP) [120].

Shear stress also triggers the development of a large outward current in atrial myocytes, shortening the atrial AP [121]. Several arguments indicate that shear-stress-induced outward potassium is due to the recruitment of K_V_1.5 channels from subcellular compartments to the sarcolemma. Notably, the delivery of K_V_1.5 channels has been directly visualized by total internal reflection fluorescence (TIRF) microscopy in live atrial cardiomyocytes submitted to laminar shear stress [121]. As for the recruitment of K_V_1.5 channels upon cholesterol depletion, the donor compartment is the recycling endosome and occurs in the same time frame, i.e., a few minutes. The exocytosis process is regulated by integrin signaling through the activation of the focal adhesion kinase (FAK) and relies on an intact microtubule system [121]. This study was the first one to give evidence that mechanical stress increase in the atrial myocardium triggers the recruitment of a pool of K_V_1.5 from an inducible reservoir. During atrial dilation caused by a hemodynamic overload, this phenomenon is exacerbated, resulting in an increased *I_Kur_* despite reduced K_V_1.5 expression levels in hypertrophied atrial myocytes. The resulting shortening of the AP could contribute to high susceptibility to developing AF in hemodynamically overloaded and dilated atria [121] (Figure 2C). Note that laminar shear stress also increases the *hERG1*-encoded current, an effect suppressed by FAK inhibitor [122]. Mutated hERG channels associated with some LQTS are insensitive to laminar shear stress [122].

During AF, drastic changes in the trafficking equilibrium as well as an alteration in the polymerization state of microtubules are observed, leading to reduced endocytosis and increased recycling [103] (Figure 2C). Taken together, this finding of up-regulated recycling pathways and reduced clathrin-mediated endocytosis is consistent with the accumulation of K_V_1.5 channels in the plasma membrane of atrial myocytes during AF.

Evidence of the role of the recycling of K_V_1.5 during atrial arrhythmias is also provided by the study of the mechanism linking mutation in *KCNE* and the risk of AF. KCNE subunits are single transmembrane domain proteins that form complexes with voltage-gated potassium (K_V_) channel α-subunits that can regulate channel trafficking. Deletion of KCNE3 is associated with susceptibility to AF in mice [123]. At the cellular level, the K_V_1.5-encoded repolarizing atrial current is enhanced due to increased Rab4-, Rab5-, and Rab9-dependent recycling of Kv1.5 channels to the z-disc/t-tubule region and LM via activation of the Akt/AS160 pathway [123]. Finally, survival of Purkinje cells following myocardial infraction depends on the subcellular remodeling of both SAP97 and the actin-binding protein cortactin, thereby preserving K_V_1.5 channel function [124].

## 5. Conclusions

Today, abundant literature has established that the trafficking of ion channels is a central regulator of normal cardiac electrophysiology and that alterations in this process are often a major cause of both inherited and acquired cardiac arrhythmias. This new knowledge on cardiac electrophysiology has profoundly transformed our view on the pathology of cardiac arrhythmias, which has shifted from a classical scheme of cardiac arrhythmia being a fixed molecular or tissue defect to its being a dynamic substrate regulated by a number of different factors. For instance, remodeling of the myocyte microarchitecture, accumulation of extracellular matrix, proliferation of non-cardiomyocyte cells, and alteration in the metabolic or oxidative status impact channel trafficking and, indeed, turn out to be important newly identified players in cardiac arrhythmias. In this light, it is easy to understand why conventional antiarrhythmic therapies targeting channel pores have demonstrated limited efficiency and why new antiarrhythmic strategies that also target cardiovascular risk factors, cardiac hemodynamic, or diet are now being investigated. A new chapter on cardiac electrophysiology and basic cardiac arrhythmias is now opened.

## Figures and Tables

**Figure 1 cells-10-02417-f001:**
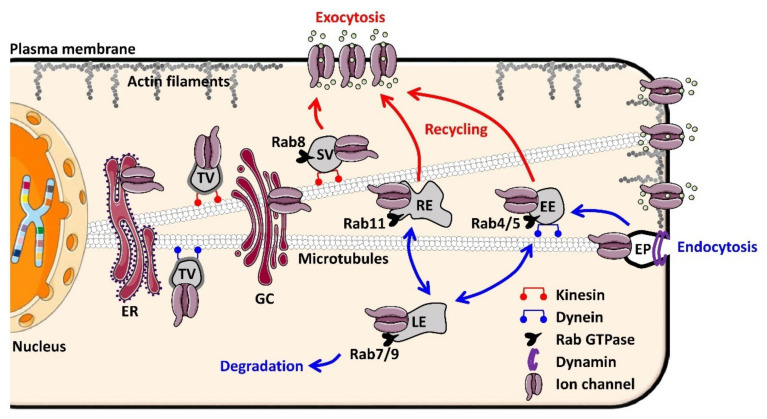
General diagram of the different steps and regulators involved in ion channel trafficking. Ion channels are directed to the plasma membrane via the anterograde and recycling pathways (red arrows). Due to their topology, proteins with multiple transmembrane domains, such as ion channels, have their *C*- and *N*-terminal domains exposed on the cytoplasmic side during transport. Endocytosis is ensured by membrane bending and fission of the endocytosis pit, mainly by dynamin (purple). Internalization signals (blue arrows) lead to degradation or recycling. Rab GTPases regulate specific steps of trafficking in the cytoplasm. Molecular motors, such as kinesin and dynein, ensure the transport of vesicles along microtubules, respectively, in the anterograde and retrograde directions. The molecular motors involved in transport along actin filaments are myosins: myosin V in the anterograde direction, myosin VI in the retrograde direction. Abbreviations: EE: early endosome; EP: endocytosis pit; ER: endoplasmic reticulum; LE: late endosome; GC: Golgi complex; RE: recycling endosome; SV: secretory vesicles (from neo-synthesis); TV: transition vesicles (shuttling between ER and the Golgi complex).

**Figure 2 cells-10-02417-f002:**
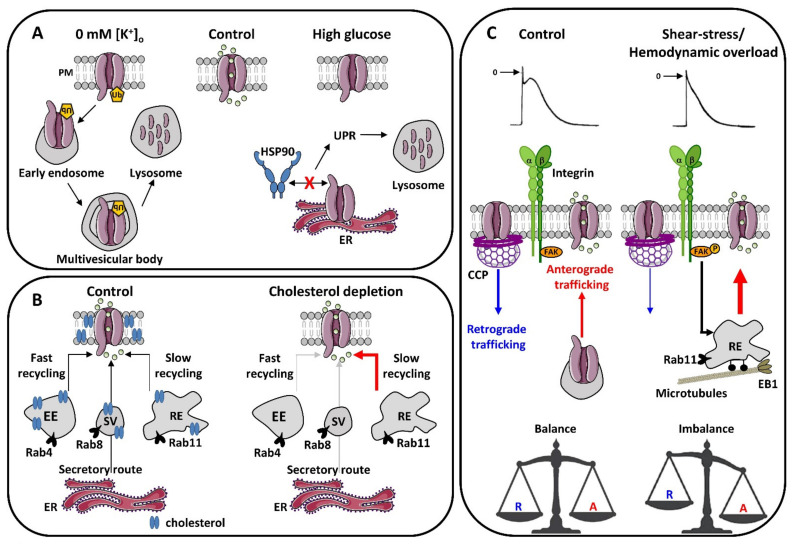
Summary diagram of alterations in the intracellular trafficking of cardiac potassium channels involved in acquired arrhythmias. (**A**) Regulation of hERG channel trafficking by hypokalemia and hyperglycemia. Decreased extracellular potassium triggers hERG channel ubiquitination, internalization, and degradation. Hyperglycemia interrupts the interaction between hERG and the chaperone HSP90 and prevents its anterograde trafficking to the membrane, activating the UPR and channel degradation. (**B**) Regulation of K_V_1.5 trafficking by cholesterol. Cholesterol depletion activates the recycling and exocytosis of the K_V_1.5 channel from the recycling endosome without affecting either the secretory route or the fast recycling pathway mediated by the early endosome. (**C**) Regulation of K_V_1.5 trafficking by mechanical constraint and atrial remodeling. In control condition, both anterograde and retrograde pathways are at equilibrium. Laminar shear stress stimulates the integrin/phospho-FAK signaling, which activates the recycling of K_V_1.5 channels accumulated in the recycling endosome. In remodeled atria and during AF, this recycling pathway is constitutively activated and the trafficking balance is altered in the direction of a favored anterograde traffic to the detriment of retrograde traffic. Abbreviations: A: anterograde; CCP: clathrin-coated pit (cleaved by dynamin); EB1: end-binding protein 1; EE: early endosome; ER: endoplasmic reticulum; FAK: focal adhesion kinase; HSP90: heat shock protein 90; P: phosphorylation; PM: plasma membrane; RE: recycling endosome; R: retrograde; SV: secretory vesicle; Ub: ubiquitylation; UPR: unfolded protein response.

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
