# Peer review of "Remodeling of Ion Channel Trafficking and Cardiac Arrhythmias"

_cells, 2021, doi:10.3390/cells10092417_

Round 1

Reviewer 1 Report

This is a very interesting review focusing on the trafficking of ion channels and arrhythmogenesis in cardiomyocytes. The authors discussed the the basic ion channel trafficking concept and how the studies showed the arrhythmias could associated with ion channel trafficking. Comments: 1. The authors discussed a lot on the anchoring part of the ion channels including Ank proteins, caveolin proteins. The reviewer is wondering whether it is clear that the arrhythmogenesis is due to the trafficking process or due to the disruption of the microdomain they locate and the alteration of the signaling regulation of such ion channels which would not be part of the trafficking of such ion channels? 2. It would be great if the authors could also discuss that the remodeling of ion channel trafficking inside the cells while trafficking within the vesicles that is associated with arrhythmogenesis. 3. Is the trafficking major affecting the function of ion channels or just affecting their location in the membrane which induce arrhythmias? 4. line 387, extra space after necessary.

Author Response

First, we are grateful for the time taken by the Editors and Reviewers in reviewing our manuscript and we greatly appreciate the opportunity to provide a responsive resubmission.

We thank the reviewer for her/his positive comment and feedback. We did our very best to address the questions and requests raised.

Below are the point-by-point answers to the Reviewer’ comments.

  1. The authors discussed a lot on the anchoring part of the ion channels including Ank proteins, caveolin proteins. The reviewer is wondering whether it is clear that the arrhythmogenesis is due to the trafficking process or due to the disruption of the microdomain they locate and the alteration of the signaling regulation of such ion channels which would not be part of the trafficking of such ion channels?

We understand the reviewer's questioning of the distinction between pure mechanisms regulating intracellular trafficking of ion channels and mechanisms regulating their organization into microdomains where ion channels are coupled to various signaling pathways.

Concerning caveolins, we consider that there is no intracellular trafficking as such since there is no evidence that caveolin-coated vesicles actively participate in ion channel transport. However, insofar as any compartment involved in the submembrane storage of ion channels (and thus non-functional since not inserted in the plasma membrane) can be recruited under the effect of, for example, adrenergic stimulation, and promote the insertion of channels into the sarcolemma and thus their functional expression, this does correspond to a trafficking process. In this sense, we consider that we can refer to intracellular trafficking. Accordingly, we have modified our text and made a distinction between what we consider to be trafficking and signaling regulation (track change in red: Page 6, lines 280-282+ Page 7, line 283, Page 7, lines 295-297). 

The paragraph on sodium channel regulation by ankyrin G has also been modified (track change in red: Page 8, lines 352-354).

  1. It would be great if the authors could also discuss that the remodeling of ion channel trafficking inside the cells while trafficking within the vesicles that is associated with arrhythmogenesis.

The reviewer's comment is extremely relevant. According to our knowledge of the current literature, a dichotomy exists between studies on trafficking that report surface regulation (availability of channels on the surface of the sarcolemma) and local regulation (correct localization of channels in specific membrane domains like DI, t-tubule). From our point of view, this dichotomy is due to the fact that it is extremely difficult to reconcile the study of trafficking mechanisms within intracellular transport vesicles and the study of the final addressing of channels in the membrane compartments of cardiomyocytes. Indeed, studies that have focused on surface regulation have used tools to manipulate intracellular organelles (ER, Golgi, endosomes) to the detriment of maintaining the structural organization of cardiomyocytes. Studies that have focused on channel targeting in specialized domains have mostly used animal models in which a partner of interest was invalidated, after which channel localization was studied in a static context (presence or absence of the channel in the specialized domains of the myocyte). A real challenge therefore lies on the reconciliation of intracellular trafficking in real time in the native environment of the cardiomyocyte. This is a major goal of our team and we hope to answer this question soon.

This point was raised in the revised version of our manuscript (track change in red: Page 2, lines 47-49): “Note that in the general concept of trafficking we will include intracellular vesicular trafficking (i.e. surface regulation) and addressing in specialized areas of the myocyte (i.e. local regulation).”

  1. Is the trafficking major affecting the function of ion channels or just affecting their location in the membrane which induce arrhythmias?

As with the reviewer's previous comment, the answer to this question is not straightforward. The subcellular localization of voltage-gated potassium channels, such as hERG and KVLQT1, has not been extensively studied to date. Therefore, it is globally the surface expression of these channels that has been shown to be involved in arrhythmias. Indeed, it is the sum of the anterograde and retrograde trafficking processes that will determine the density of functional channels present in the sarcolemma, and thus the duration of the AP. Similarly, the subcellular localization of inward-rectifiers remains poorly understood, unless one considers that Kir2.x and NaV1.5 are part of a macrocomplex and that the localization of Kir channels must therefore follow that of NaV1.5 channels. However, this is still speculative as it has not been demonstrated.

Conversely, the localization of the NaV1.5 sodium channel has been and remains widely studied. An abundant literature using KO animal models for different partners of the channel reports that if the channel is not localized in the expected domains, it is because there is a trafficking defect that leads to a targeting defect. We consider that this vision is perhaps a little simplistic because the dynamic dimension cannot be apprehended in these models. Indeed, if it is possible to consider an anterograde traffic defect, it is not possible to exclude an interaction and stabilization defect of the channel leading to its internalization. Therefore, there is an urgent need today to be able to follow the dynamic trafficking of the channel and its targeting towards the microdomains of native cardiomyocytes. Ideally, to model the mechanisms involved in cardiac arrhythmias, the use of hiPSC-derived CMs from patient would be very relevant. However, the lack of maturity of these cells is currently not suitable for studying targeting in microdomains. 

  1. line 387, extra space after necessary.

This has been corrected. In addition, we used MDPI's English Editing service before resubmitting the manuscript (track changes in purple and orange).

Reviewer 2 Report

This is a thorough review regarding the roles of altered ion channel trafficking in generation of cardiac arrhythmias.  The discussion is well organized and will be of interest to individuals in the cardiovascular and ion channel fields.  There are numerous minor grammatical issues that need to be carefully edited.  In my opinion, a brief introduction to cardiac ion channels and/or a schematic illustrating the localization of different channels (atria versus ventricular, etc) may be beneficial to investigators not in the cardiac ion channel field.

Author Response

First, we greatly appreciate the opportunity to provide a responsive resubmission. We are grateful for the time taken by the Reviewer in reviewing our manuscript.

We thank the reviewer for her/his positive comment and feedback. Below are the point-by-point answers to the Reviewer’ comments.

  1. There are numerous minor grammatical issues that need to be carefully edited. 

On the reviewer suggestion, we used MDPI's English Editing service before resubmitting the manuscript (track changes in purple and orange).

  1. A brief introduction to cardiac ion channels and/or a schematic illustrating the localization of different channels (atria versus ventricular, etc) may be beneficial to investigators not in the cardiac ion channel field.

As requested by the Reviewer, we provide a brief introduction to cardiac ion channels (track change in red: Page 2, Lines 52-74) and an additional Reference ([1]).

We are afraid that an exhaustive review of the literature summarizing the location of all channels involved in atrial and ventricular AP is hardly feasible within the 5 days allowed for the revision of the manuscript. However, this information is available in our previous publication (Balse et al., Physiol. Rev. 2012, Reference [95]).